# Partial Sequence Analysis of Commercial Peste des Petits Ruminants Vaccines Produced in Africa

**DOI:** 10.3390/vetsci11100500

**Published:** 2024-10-13

**Authors:** Boubacar Barry, Yebechaye Tessema, Hassen Gelaw, Cisse Rahamatou Moustapha Boukary, Baziki Jean de Dieu, Melesse Ayelet Gelagay, Ethel Chitsungo, Richard Rayson Sanga, Gbolahanmi Akinola Oladosu, Nick Nwankpa, S. Charles Bodjo

**Affiliations:** 1Pan African University Life and Earth Sciences Institute (Including Health and Agriculture), Ibadan 200284, Oyo State, Nigeria; bbarry083@gmail.com (B.B.); richardrayson94@gmail.com (R.R.S.); oladosugbolahanmi@gmail.com (G.A.O.); 2African Union-Pan African Veterinary Vaccine Centre (AU-PANVAC), Debre Zeit P.O. Box 1746, Ethiopia; yebechayet@africa-union.org (Y.T.); hasseng@africa-union.org (H.G.); boukaryc@africa-union.org (C.R.M.B.); bazikij@africa-union.org (B.J.d.D.); gelagaya@africa-union.org (M.A.G.); ethelc@africa-union.org (E.C.); nicknwankpa@gmail.com (N.N.)

**Keywords:** peste des petits ruminants, vaccine, genetic stability, molecular evolution, master seed

## Abstract

**Simple Summary:**

Peste des petits ruminants (PPR) is a highly contagious viral disease affecting small ruminants, posing significant economic challenges, especially in Africa. This study focused on understanding the genetic stability of the live-attenuated Peste des petits ruminant’s vaccine Nigeria 75/1 strain, which is an essential tool for the eradication program. A nucleotide sequence comparison of the hypervariable region (C-terminus domain) of the nucleoprotein of the PPR vaccine from 10 different African vaccine manufacturers was conducted. Data analysis revealed 100% nucleotide homology between vaccine samples and the master seed, indicating the genetic stability of the PPR vaccine Nigeria 75/1 over decades. This research contributes to ongoing efforts to control Peste des petits ruminants outbreaks and ensures the efficacy of vaccination programs, ultimately benefiting animal health and agricultural sustainability.

**Abstract:**

Peste des petits ruminants virus (PPRV), which is the only member of the *Morbillivirus caprinae* species and belongs to the genus *Morbillivirus* within the *Paramyxoviridae* family, causes the highly contagious viral sickness “Peste des petits ruminants (PPR).” PPR is of serious economic significance for small ruminant production, particularly in Africa. Control of this critical disease depends highly on successful vaccination against the PPRV. An in-depth understanding of the genetic evolution of the live-attenuated PPR vaccine Nigeria 75/1 strain used in Africa is essential for the successful eradication of this disease by 2030. Therefore, this study investigated the possible genetic evolution of the PPR vaccine produced by various African laboratories compared with the master seed available at AU-PANVAC. RT-PCR was performed to amplify a segment of the hypervariable C-terminal part of the nucleoprotein (N) from commercial batches of PPR vaccine Nigeria 75/1 strain. The sequences were analyzed, and 100% nucleotide sequence identity was observed between the master seed and vaccines produced. The results of this study indicate the genetic stability of the PPR vaccine from the Nigeria 75/1 strain over decades and that the vaccine production process used by different manufacturers did not contribute to the emergence of mutations in the vaccine strain.

## 1. Introduction

Peste des petits ruminants (PPR) is a highly contagious viral disease affecting small domestic (sheep and goats) and wild ruminants [1,2,3]. Although PPR was initially reported in Cote d’Ivoire in West Africa in 1942 [4], the geographic distribution of the disease has expanded far from West Africa. The disease has now been noticed in over 70 African nations, the Middle East, and Asia [5]. Over 80% of the global small ruminant population is in these areas, and over 330 million people depend on small ruminants [5]. PPR is an economically important animal illness with a significant impact on sheep and goat production with high morbidity and mortality rates, particularly in the naïve populations [6]. The annual global repercussions are estimated to range from USD 1.4 billion to USD 2.1 billion [7]. Due to the substantial influence of this disease on the production of small ruminants, PPR is presently recognized internationally as a threat to the sheep and goat populations. The disease has been chosen for eradication by 2030 with the developed PPR Global Control and Eradication Strategy (PPR-GCES) [8].

PPR is caused by the peste des petits ruminants virus (PPRV), the only member of the *Morbillivirus caprinae* species, which belongs to the genus *Morbillivirus* within the *Paramyxoviridae* family since the revision of virus nomenclature by the International Committee on Taxonomy of Viruses (ICTV) [9,10]. The genome of PPRV is comprised of 15,948 nucleotides. It contains six genes arranged in the 3′-N-P-M-F-H-L-5′ orientation, which encode six polypeptides: nucleocapsid protein (N), phosphoprotein (P), matrix protein (M), fusion protein (F), hemagglutinin (H), and large RNA-dependent polymerase protein (L). The virus is a negative-sense, linear, single-stranded, and non-segmented RNA virus. PPRV exists as a single serotype, but partial sequences of the C-terminus of the nucleoprotein gene (N) and the fusion protein gene (F) allow the identification of four genetically distinct lineages: I, II, III, and IV [11,12,13,14]. The C-terminal region of the nucleoprotein N gene is the most variable region of this protein among morbilliviruses [15].

The strategy for eradicating PPR in infected countries is based on vaccination campaigns using two live attenuated vaccines—the Nigeria 75/1 and Sungri/96 strains—belonging to lineages II and IV, respectively [16,17]. The Nigeria 75/1 vaccine, developed in 1989, is extensively produced and the most widely used strain globally. It was obtained after attenuation of a wild strain following 75 passages in cell culture. Sequence comparisons between the original wild-type virus and the Nigeria 75/1 vaccine strain showed that 18 fixed mutations separate the two strains, among which two are in the nucleoprotein gene [18]. The Nigeria 75/1 vaccine has been extensively produced by African vaccine manufacturers over the years. It is recommended that manufacturers do not make a passage of the master seed more than 10 times for the production of final batches of PPR vaccines [19] to avoid the generation of viral variants and possible reversion to virulence. Determining the genetic stability of the PPR vaccine Nigeria 75/1 strain produced is crucial to ensure the success of the ongoing Global Eradication Program. The current study investigated the possible introduction of mutations over time in the PPR vaccine Nigeria 75/1 strain produced by African vaccine manufacturers.

## 2. Materials and Methods

### 2.1. PPR Vaccine Samples

Forty-four vials of PPR vaccine Nigerian 75/1 strain from ten African vaccine manufacturers produced from 2017 to 2022 sent to AU-PANVAC for quality control and stored at −20 °C were selected (Table 1). Most of these vaccine manufacturers received the PPR vaccine seed from AU-PANVAC, except MCI-Santé Animal of Morocco, who received the seed directly from CIRAD-France. One vial per batch of all these vaccines has passed the identity test carried out for quality control purposes and has been randomly selected regardless of expiry date, ensuring a balanced representation across manufacturers. The PPR vaccine seed and a known PPR vaccine used as a positive control (PC) in the titration available at AU-PANVAC were also included in the study.

### 2.2. Viral Genome Extraction

The viral nucleic acid (RNA) was extracted from the freeze-dried vaccine vials reconstituted with 2 mL of PBS (pH = 7.2) using the QIAamp^®^ Viral RNA Mini Kit (Qiagen^®^ QIAamp Viral RNA Mini Kit Cat no. 52906, Hilden, Germany), adhering to the manufacturer’s instructions and AU-PANVAC SOPs. The extracted RNA from PPR vaccines was shipped to the Animal Production and Health Laboratory (APHL) of the Joint FAO/IAEA Division, Department of Nuclear Sciences and Applications, International Atomic Energy Agency (Vienna, Austria).

### 2.3. Reverse Transcription Polymerase Chain Reaction (RT-PCR) and Nucleotide Sequences Analysis

At the APHL, amplifications of 351 base-pair (bp) fragments of the hypervariable C-terminus of the N gene were conducted with the primers NP3 5′-GTC-TCG-GAA-ATC-GCC-TCA-CAG-ACT-3′ and NP4 5′-CCT-CCT-CCT-GGT-CCT-CCA-GAA-TCT-3′ [20]. One-step RT-PCR was performed using the kit according to the manufacturer’s instructions (Qiagen Cat No: 210212, Hilden, Germany). In short, 5 µL of RNA extract or RNase-free water for the negative control (NC) was used as template to be amplified in a 25 µL reaction mix containing 8 µL RNase-free water, 5 µL of one-step RT-PCR Buffer 5X, 1 µL of dNTP 10 mM mix each, 2.5 µL of each primer (NP3 and NP4) at a final concentration of 0.5 µM, 1 µL of one-step RT-PCR Mix. The thermal cycler (BIO-RAD C1000 Touch Thermal cycler, Hercules, CA 94547, USA) conditions were as follows: reverse transcription for 30 min at 50 °C, initial PCR activation for 5 min at 94 °C, followed by 30 cycles of amplification corresponding to 30 s at 94 °C, 30 s at 55 °C, 30 s at 72 °C, and a final extension for 5 min at 72 °C. RT-PCR expected an amplification product of 351 bp. Five microliters of the RT-PCR amplified products were mixed with gel-red nucleic acid gel stain (BIOTIUM Cat. No: BTM41003-1) and analyzed in gel electrophoresis after one hour, running at 100 volts. The purified positive RT-PCR amplicons were sent to LGC Genomics (Berlin, Germany) for sequencing using the same type of primers (NP3 and NP4).

The sequences generated were assembled using the Staden Package (http://staden.sourceforge.net/, accessed on 10 October 2023). The nucleotides sequence length of 287 bp corresponding to the partial amino-acid sequence of the less-conserved carboxy-terminal region of N protein was analyzed, and sequence alignment was performed using MUSCLE (https://www.ebi.ac.uk/jdispatcher/msa/muscle, accessed on 22 September 2024) [21]. The raw data obtained were analyzed using a bioinformatics pipeline to determine the consensus sequence within the sample and detect nucleotide variations (SNVs) within the study vaccine samples.

### 2.4. Sequence Analysis with Published Sequence or in Genbank of PPR Vaccine Nigeria 75/1

The consensus sequence obtained from the analyzed PPR vaccines was aligned with the sequences from vaccine strain Nigeria 75/1 genes already published [15] or in Genbank (Accession KY628761) as indicated above.

## 3. Results

### 3.1. Detection and Identification of the PPRV N Gene Using PCR

The amplified gene segment of the C-terminus of the N gene with primers NP3 and NP4 were detected by electrophoresis in agarose gel, which revealed the successful detection of the amplicons of 351 bp, as previously described [20] (see Figure 1). Except sample number 27, all other samples were found positive. The AU-PANVAC PPR vaccine positive control (PC), used as the study’s PC, was also amplified.

### 3.2. Sequencing and Data Analysis

The forty-four (45) sequences of 287 nucleotides were generated and analyzed using MUSCLE [21]. Sequence alignment showed 100% identity among PPR vaccine samples and the seed vial (Figure 2a,b). The alignment of two published sequences of Nigeria 75/1 strain (Genbank KY628761 and Diallo, 1994) with our generated PPR consensus sequence of the 287 bp showed a single nucleotide polymorphism change (Figure 3), where the nucleotide adenine “A” at position 1448 of the N gene is substituted by the nucleotide cytosine “C”.

## 4. Discussion

PPR has spread far beyond its Western African origin over the last decades. Due to its significant impact on sheep and goat production in the affected areas, and taking lessons learned from the successful eradication of Rinderpest, PPR has been targeted for eradication by 2030 [4]. As for Rinderpest, the eradication strategy is mainly based on extensive vaccination using currently available live attenuated PPR vaccines [16,17].

Since its development, the Nigeria 75/1 PPR vaccine strain has been extensively used over the years in different parts of the world. In Africa, the production of a homologous PPR vaccine with the Nigeria 75/1 strain by manufacturers started more than 30 years ago. As the continental PPR eradication strategy will focus mainly on vaccination, it is important to understand any possible molecular evolution of PPR vaccine Nigeria 75/1 strain, which could impact the success of the eradication program.

This current study analyzed the partial sequence of the N gene in the C-terminus region from forty-three PPR vaccine batches produced by African vaccine manufacturers, the PPR vaccine seed, and a PPR vaccine produced at AU-PANVAC. The negative result for sample 27 could be due to RNA degradation during the shipment, as this sample was confirmed by RT-PCR performed at AU-PANVAC. Alignment of the 287-bp sequence in the C-terminus region, corresponding to nucleotides 1283 to 1570 of the N gene, showed 100% identity between all the sequences generated. This seems to indicate a genetic stability of the PPR vaccine using the Nigeria 75/1 strain produced in Africa. Indeed, the C-terminus region of the N gene is the least conserved among morbillivirus [14] and is used for PPRV lineage differentiation.

However, the alignment of two published sequences of Nigeria 75/1 strain (Genbank KY628761 and Diallo, 1994) with our generated PPR consensus sequence of the 287 bp showed a single nucleotide polymorphism change, where the nucleotide adenine “A” at position 1448 of the N gene is substituted by the nucleotide cytosine “C”. This change also impacts the amino-acid sequence, as the Glutamine (Q) at position 476 (codon CAA) in two published sequences of Nigeria 75/1 strain (Genbank KY628761 and Diallo, 1994) is replaced by Proline (P) with codon CCA in our generated PPR consensus sequence. This might be due to the accuracy of the sequencing process, as all manufacturers in the study received the same PPRV Nigeria 75/1 vaccine master seed in AU-PANVAC, initially originated by CIRAD.

Our finding bodes well for the potential effectiveness and uniformity of the vaccines, offering valuable insights for further research and development initiatives. The whole-genome sequencing approach would offer a more comprehensive understanding, allowing for a detailed examination of the genetic makeup beyond the confines of the relatively short C-terminus hypervariable region. Exploring the entire genome can provide a comprehensive understanding of potential deviations and variations, enhancing the robustness of genetic analysis and providing a more nuanced perspective on the PPRV vaccine’s molecular evolution.

In conclusion, the analysis conducted in this study revealed remarkable uniformity among all Peste des petits ruminants virus (PPRV) vaccine samples, demonstrating a sequence identity of 100% among them and with the master seed. These findings advance our understanding of the PPRV vaccine genetic landscape, offering practical implications for vaccination using the Nigeria 75/1 strain.

## Figures and Tables

**Figure 1 vetsci-11-00500-f001:**
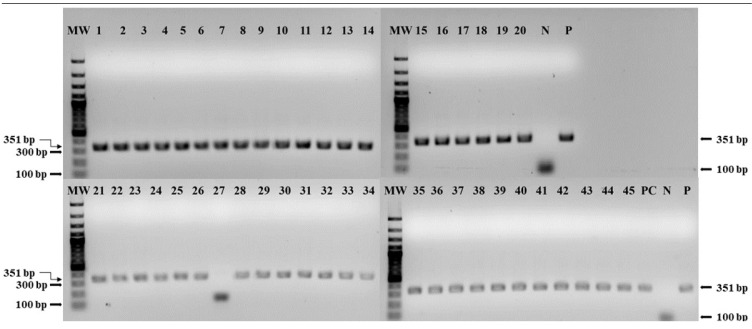
Gel electrophoresis results of RT-PCR with PPR vaccine samples (1 to 45) from 10 Manufacturers, (1–10) from Manufacturer 01 (NVI), (11–12) from Manufacturer 02 (KEVEVAPI), (13–20) from Manufacturer 03 (LANAVET), (21–24) from Manufacturer 04 (MCI), (25–30) from Manufacturer 05 (ISRA), (31–37) from Manufacturer 06 (LCV), (38–40) from Manufacturer 07 (BVI), (41–42) from Manufacturer 08 (HESTER Tanzania), (43) from Manufacturer 09 (NVRI), (44) from Manufacturer 10 (NAPHL), and (45) the master seed (master seed, Vero 78), MW: 100 bp molecular weight marker, PC: AU-PANVAC PPR vaccine positive control, P: positive control, and N: negative control. The required band amplification with primers NP3 and NP4 is at 351 bp.

**Figure 2 vetsci-11-00500-f002:**
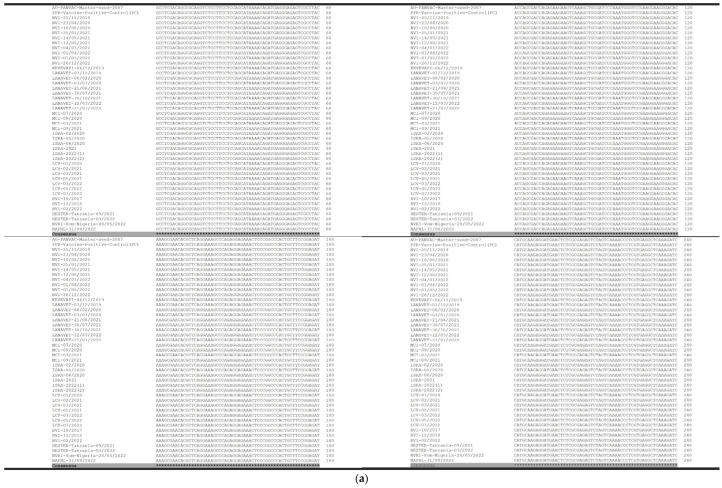
(**a**) Multiple sequence alignment of 287-bp sequence in the C-terminus region, corresponding to nucleotides 1283 to 1522 of the N gene generated with PPR vaccine samples using Clustal Omega < EMBL-EBI. Data have shown sequence homology of 100%. (**b**) Multiple sequence alignment of 287-bp sequence in the C-terminus region, corresponding to nucleotides 1253 to 1570 of the N gene generated with PPR vaccine samples using Clustal Omega < EMBL-EBI. Data have shown a sequence homology of 100.

**Figure 3 vetsci-11-00500-f003:**
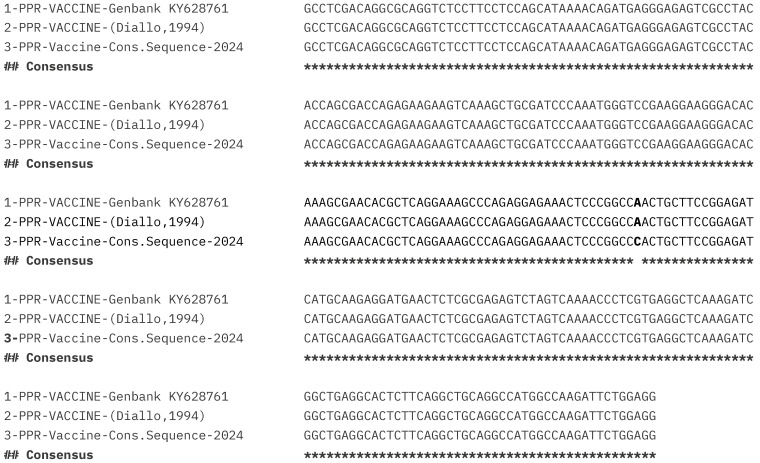
Alignment of the generated consensus partial sequence of N gene (287 bp: nucleotides 1283 to 1448) with two published sequences of PPR virus Nigeria 75/1 strain (Genbank KY628761 and Diallo, 1994) using Clustal Omega < EMBL-EBI.

**Table 1 vetsci-11-00500-t001:** Selected vaccine samples available to AU-PANVAC from different African PPR manufacturers for quality control from 2020 to 2023. AU-PANVAC’s known PPR vaccine, which passed the identity test, was utilized as the study’s positive control (PC). The seed is from May 2007.

S/N	Manufacturer	Production Date
1	NVI	11/2019
2	NVI	04/2020
3	NVI	08/2020
4	NVI	01/2021
5	NVI	05/2021
6	NVI	08/2021
7	NVI	01/2022
8	NVI	04/2022
9	NVI	05/2022
10	NVI	12/2022
11	KEVEVAPI	12/2019
12	KEVEVAPI	12/2019
13	LANAVET	12/2019
14	LANAVET	02/2020
15	LANAVET	07/2020
16	LANAVET	06/2021
17	LANAVET	07/2021
18	LANAVET	10/2021
19	LANAVET	03/2022
20	LANAVET	07/2022
21	MCI	07/2020
22	MCI	09/2020
23	MCI	03/2021
24	MCI	09/2021
25	ISRA	02/2020
26	ISRA	05/2020
27	ISRA	06/2020
28	ISRA	10/2021
29	ISRA	05/2022
30	ISRA	11/2022
31	LCV	01/2020
32	LCV	02/2021
33	LCV	03/2021
34	LCV	05/2021
35	LCV	03/2022
36	LCV	05/2022
37	LCV	07/2022
38	BVI	10/2017
39	BVI	12/2018
40	BVI	02/2022
41	HESTER Tanzania	09/2021
42	HESTER Tanzania	03/2022
43	NVRI	05/2022
44	NAPHL	08/2022
45	AU-PANVAC (Master seed, Vero 78)	05/2007
46	AU-PANVAC PPR Vaccine Positive Control (PC)	01/2022

## Data Availability

Data available in the AU-PANVAC archives.

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
