# Peer review of "Partial Sequence Analysis of Commercial Peste des Petits Ruminants Vaccines Produced in Africa"

_vetsci, 2024, doi:10.3390/vetsci11100500_

Round 1

Reviewer 1 Report

Comments and Suggestions for Authors

This study compared sequences of PPR virus in commercial vaccines with the Master seed. No genetic evolution was found in vaccine strains. This finding ensures that the vaccines are still effective.

·  General comments 

Introduction

-Previous reports on phylogenetic analysis of wild type strains of the virus should be included to provide basic information about variation of the virus.

Materials and methods

-Detail about sampling should be clearly described to declare selection bias.

Discussion

-It would be interesting if you compare your results with previous studies on genetic stability of vaccine strain PPR virus or other related viruses.

-Could you please discuss about efficacy of vaccines based on your finding and previous reports on phylogenetic analysis of wild type strains of the virus. 

·  Specific comments 

The beginning of the legend of figure 1 may be missed.

Please clearly define lane 1-45 in figure 1.

The end of legend of figure 3 may be missed.

Please reorganize table 1.

Reference [15] was not cited in the body of the manuscript.

Please recheck format of references.

Author Response

RESPONSE TO REVIEWER 1 (Vetsci-3154442)

Comments and Suggestions for Authors

This study compared sequences of PPR virus in commercial vaccines with the Master seed. No genetic evolution was found in vaccine strains. This finding ensures that the vaccines are still effective.

General comments raised

Materials and methods

-Detail about sampling should be clearly described to declare selection bias.

Authors Response/Action

Correction has been made.

Specific comments raised

The beginning of the legend of figure 1 may be missed.

Authors Response/Action

Correction has been made.

Please clearly define lane 1-45 in figure 1.

Authors Response/Action

Correction has been made.

The end of legend of figure 3 may be missed.

Authors Response/Action

Correction has been made.

Please reorganize table 1.

Authors Response/Action

Correction has been made.

Reference [15] was not cited in the body of the manuscript.

Authors Response/Action

Correction has been made.

Please recheck format of references

Authors Response/Action

Correction has been made.

Reviewer 2 Report

Comments and Suggestions for Authors

The authors sequenced the nucleic acid sequence of the C-terminal hypervariable region of the N protein for several batches of commercialized PPRV vaccines used in Africa in recent years, and concluded that the genetic stability of the corporate-produced PPRV vaccines is reliable. However, PPRV, as an RNA virus, has a very high probability of producing sequence mutations during amplification. The authors only performed the analysis of the sequence of the highly variable region of the N protein, which is not quite enough to determine its genetic stability. The authors need to refer to the sequence characterization of PPRV virulence proteins and add information to verify the mutations in the key sites of some virulence proteins. Alternatively, they should elaborate on the technical protocols used by companies to ensure the genetic stability of the virus in their protocols for the production of PPRV and compare them with the protocols used in this study, and explore how the studies in this manuscript support the conclusions.

Author Response

RESPONSE TO REVIEWER 2 (Vetsci-3154442)

Comments and Suggestions raised

The authors sequenced the nucleic acid sequence of the C-terminal hypervariable region of the N protein for several batches of commercialized PPRV vaccines used in Africa in recent years, and concluded that the genetic stability of the corporate-produced PPRV vaccines is reliable. However, PPRV, as an RNA virus, has a very high probability of producing sequence mutations during amplification. The authors only performed the analysis of the sequence of the highly variable region of the N protein, which is not quite enough to determine its genetic stability. The authors need to refer to the sequence characterization of PPRV virulence proteins and add information to verify the mutations in the key sites of some virulence proteins. Alternatively, they should elaborate on the technical protocols used by companies to ensure the genetic stability of the virus in their protocols for the production of PPRV and compare them with the protocols used in this study, and explore how the studies in this manuscript support the conclusions.

Authors Response/Action

Alignment of morbillivirus N proteins defined four regions with varying degrees of homology. Region. Ther C terminus region comprising the amplicon of 351bp is the least conserved with 30.5%, 26-6%, 18"4% and 17.4% among morbillivirus which is also used for PPR virus lineage determination. [Diallo, A., et al. (1995): "Cloning of the nucleocapsid protein gene of Peste-des-petits-ruminants virus: relationship to other members of the Morbillivirus genus." Journal of General Virology, 76(10), 2331-2339.]

This preliminary work will continue later with the whole-genome sequencing approach to provide a comprehensive understanding of potential deviations and mutation on the PPR vaccine strain.

Reviewer 3 Report

Comments and Suggestions for Authors

My full review is attached as a .pdf.

Comments on the Quality of English Language

The manuscript is very well written but could use some polishing.

Author Response

Answers for Vetsci-3154442 Review (Reviewer 3)

Why was only a partial 351bp fragment of the N gene sequenced and not the complete N gene or whole PPRV sequenced? What evidence is there that only changes in this fragment of the N gene important for vaccine efficacy or reversion from an attenuated virus to infectious virion.

             Answer:

Alignment of morbillivirus N proteins defined four regions with varying degrees of homology. Region. Ther C terminus region comprising the amplicon of 351bp is the least conserved with 30.5%, 26-6%, 18"4% and 17.4% among morbillivirus which is also used for PPR virus lineage determination. [Diallo, A., et al. (1995): "Cloning of the nucleocapsid protein gene of Peste-des-petits-ruminants virus: relationship to other members of the Morbillivirus genus." Journal of General Virology, 76(10), 2331-2339.]

This preliminary work will continue later with the whole-genome sequencing approach to provide a comprehensive understanding of potential deviations and mutation on the PPR vaccine strain.

Comment raised

Title

  • The title describes what was done but doesn’t highlight the main findings. Change the title to reveal what your study shows and why it’s important.

Authors Response/Action

We thing that the title can revised as  “Partial sequence analysis of commercial Peste des Petits Ruminants vaccines produced in Africa: Homogeneity of sequences and implications of vaccine strain stability”

Comment raised

Simple Summary

  • Remove “… by 2030.”

Authors Response/Action

Correction has been made.

Comment raised

Abstract

  • Need a period after “… within the Paramyxoviridae family”

Authors Response/Action

Correction has been made.

Comment raised

Introduction

  • “High morbidity” has a range of 10-90%. 10% is not considered high. Please explain the range or remove the numbers.
  • “… non-segmented RNA microorganism” change microorganism to virus.

Authors Response/Action

  • Correction has been made.
  • Correction has been made.

Comments raised

Materials and Methods

  • Please disclose what was used as positive and negative controls.

Authors Response/Action

  • AU-PANVAC's known PPR vaccine, which passed the identity test was utilised as the study's Positive Control (PC), while PBS (pH=7.2) was used as the Negative Control (NC) throughout.
  • “… African vaccine manufacturers and produced” – remove “and”

Authors Response/Action

Correction has been made.

  • Section 2.3 RT-PCR section
    • What do you mean by “partial amplification”? The PCR obviously amplified the fragment.

Authors Response/Action

As we only amplified the 351 base-pair (bp) fragments of the C-terminus of the N gene not the whole N gene made us state “partial amplification”

  • What PCR kit was used? Please provide the name and kit number.

Authors Response/Action

  • RNA extraction from samples was done using a commercial RNA extraction kit (Qiagen® QIAamp Viral RNA Mini Kit Cat no. 52906, Germany) as per the manufacturer’s instructions.
  • Reverse Transcription- Polymerase Chain Reaction (RT-PCR) was performed Qiagen One Step PCR kit (with One step RT- PCR buffer, one Step RT- PCR enzyme mix , dNTP mix, RNase free water) Cat No: 210212 was used for the PCR.
  • “5 uL of RNA extraction” should be RNA extract.

Authors Response/Action

Correction has been made.

  • A 30 min RT step at 50C seems too long. Please double check the protocol.

Answer:

Based on AU-PANVAC’s protocol and various research papers the PPR PCR thermal cycle, the Reverse transcription step stays at temperature of 500C for 30 min.

  • What thermocycler was used?

Answer:

The thermal cycler used for this study was Bio-rad C1000 Touch Thermal cycler.

  • For agarose gel electrophoresis, how long was the gel run for and at what voltage. Is this an Ethidium Bromide gel or something else?

Answer:

For Agarose gel electrophoresis, it was run for 1 hour at 100 Voltage. The Gel electrophoresis run using Gel-red Nucleic acid gel stain cat. No: 41003

  • What ladder was used? Company and number

Answer:

The DNA molecular weight marker used for the study was QIAGEN GelPilot 100 bp Plus

Ladder (100) (cat. no. 239045).

Comments raised

Results

  • In 3.1 need to clearly state that 44/45 vials amplified.

Authors Response/Action

Correction has been made.

  • As sample #27 did not amplify was the extraction and PCR repeated to ensure that the vaccine was simply not processed or handled correctly?

Authors Response/Action

No, the extraction and PCR were not repeated in . However, at the AU-PANVAC, prior to shipment, extraction and PCR were performed for all the selected samples. All the samples passed the PPR identity test.

  • The explanation about #27 being degraded needs to move from the results to the discussion section.

Authors Response/Action

Correction has been made.

  • Figure 1 needs a proper caption as it doesn’t say that this is an agarose DNA gel and what the numbers refer to. There isn’t even a proper sentence in the caption. Also move the information about #27 to the discussion section.

Authors Response/Action

Correction has been made.

  • Section 3.2 – if only 44/45 vaccine vials amplified in Section 3.1 how were 45 sequences of 287 bp sequenced? If #27 didn’t amplify than how can you sequence it?

Authors Response/Action

  • Forty-five (45) vials, from which 44 PPR vaccines from ten (10) African vaccine manufacturers and a vial of Nigerian 75/1 master seed Vero 78 in the AU-PANVAC repository produced in 2007 were selected for this study (Table 1). AU-PANVAC's known PPR vaccine, which passed the identity test and used as this study's Positive Control (PC) throughout was also added to the shipment. Everything was 46 including the PC. One (#27) of the 44 vaccines didn’t amplify. That means 43 amplified vaccines+the seed+the PC=45 sequences of 287 bp.
  • Why only sequence 287bp and not the full 351bp amplicon?

Authors Response/Action

The sequence date received from the service provider allow us to alignme a maximum of of 287bp.

  • Figure 2 is impossible to read and it is unclear why there are 4 boxes and not only 1. This figure needs to be redone so it is readable. You can also use dots to show sequence homology so that if any base pair was different, it would stand out. It is unclear the difference between Fig 2a and 2b. Only one is sufficient.

Authors Response/Action

Figure 2.a Multiple sequence alignment of Alignment of 287-bp sequence in the C-terminus region, corresponding to nucleotides 1283 to 1522 of the N gene generated with PPR vaccines samples using Clustal Omega < EMBL-EBI. Data has shown Sequence homology of 100%.

Figure 2.b Multiple sequence alignment of Alignment of 287-bp sequence in the C-terminus region, corresponding to nucleotides 1253 to 1570 of the N gene generated with PPR vaccines samples using Clustal Omega < EMBL-EBI. Data has shown Sequence homology of 100

Comments raised

Discussion

  • Information about the single nucleotide polymorphism and Figure 3 should be moved to the results section and discussed in the Discussion.

Authors Response/Action

Correction has been made.

  • Figure 3. How was the alignment made? What software was used?

Authors Response/Action

Figure 3. Alignment of the generated consensus partial sequence of N gene (287 bp: nucleotides 1283 to 1448) with two published sequences of PPR virus Nigeria 75/1 strain (Genbank KY628761 and Diallo, 1994) using Clustal Omega < EMBL-EBI.

  • Table 1 was not mentioned in the text and needs to be moved to the methods section. The table also doesn’t fit onto the page and the information in the top half of the table change halfway down and the table is hard to understand.

Authors Response/Action

Table 1 was mentioned in the text in Section 2.1. (PPR vaccine samples). Correction has been made.
